# Idiopathic Pulmonary Fibrosis and Post-COVID-19 Lung Fibrosis: Links and Risks

**DOI:** 10.3390/microorganisms11040895

**Published:** 2023-03-30

**Authors:** Filippo Patrucco, Paolo Solidoro, Francesco Gavelli, Daria Apostolo, Mattia Bellan

**Affiliations:** 1Respiratory Diseases Unit, Medical Department, AOU Maggiore della Carità Hospital, 28100 Novara, Italy; 2Medical Sciences Department, University of Turin, 10126 Turin, Italy; 3Respiratory Diseases Unit, Cardiovascular and Thoracic Department, AOU Città della Salute e della Scienza di Torino, 10126 Turin, Italy; 4Translational Medicine Department, University of Eastern Piedmont, 28100 Novara, Italy; 5Emergency Medicine Department, Maggiore della Carità Hospital, 28100 Novara, Italy; 6Division of Internal Medicine, Medical Department, AOU Maggiore della Carità di Novara, 28100 Novara, Italy

**Keywords:** COVID-19, idiopathic pulmonary fibrosis, post-COVID-19 pulmonary fibrosis, pulmonary fibrosis, SARS-CoV-2

## Abstract

Idiopathic pulmonary fibrosis (IPF) is considered the paradigmatic example of chronic progressive fibrosing disease; IPF does not result from a primary immunopathogenic mechanism, but immune cells play a complex role in orchestrating the fibrosing response. These cells are activated by pathogen-associated or danger-associated molecular patterns generating pro-fibrotic pathways or downregulating anti-fibrotic agents. Post-COVID pulmonary fibrosis (PCPF) is an emerging clinical entity, following SARS-CoV-2 infection; it shares many clinical, pathological, and immune features with IPF. Similarities between IPF and PCPF can be found in intra- and extracellular physiopathological pro-fibrotic processes, genetic signatures, as well as in the response to antifibrotic treatments. Moreover, SARS-CoV-2 infection can be a cause of acute exacerbation of IPF (AE-IPF), which can negatively impact on IPF patients’ prognosis. In this narrative review, we explore the pathophysiological aspects of IPF, with particular attention given to the intracellular signaling involved in the generation of fibrosis in IPF and during the SARS-CoV-2 infection, and the similarities between IPF and PCPF. Finally, we focus on COVID-19 and IPF in clinical practice.

## 1. Idiopathic Pulmonary Fibrosis

Idiopathic pulmonary fibrosis (IPF) is a chronic, fibrosing, interstitial pneumonia of unknown cause [1]; it is the most common type of idiopathic interstitial pneumonias and it is characterized by histopathologic and radiological features of usual interstitial pneumonia (UIP) [2]. UIP generally presents with honeycombing, traction bronchiectasis, and peripheral alveolar septal thickening [2].

The co-responsibility of three pathogenetic elements—genetic predisposition [3], environmental factors [4], and accelerated-aging-associated changes [5]—results in a complex epigenetic reprogramming, promoting the aberrant epithelial cell activation. Upon activation, epithelial cells secrete many mediators that induce migration, proliferation, and activation of fibroblasts and myofibroblasts. These cells are resistant to apoptotic mechanisms and continue to secrete extracellular matrix components [2]. Moreover, extracellular matrix is a reservoir of several growth factors that are released as soluble ligands upon degradation; these mechanisms are self-sustained by positive feedback loops and the signaling crosstalk, causing the complexity of the system and being responsible for the inexorable progression of the fibrotic disease [2].

The pathological result is the replacement of the normal compliant lung extracellular matrix with an altered one, rich in fibrillar collagen [6]. The UIP pattern of alveolar lesions is characterized by: (1) spatial heterogeneity of the lesions, alternated with normal lung areas; (2) temporal heterogeneity, with the coexistence of discrete lesions in lung tissue that appears normal (the so-called fibroblast foci) and fibrotic areas mainly constituted by a dense acellular collage; (3) honeycombing lesions, which are dilated airspaces with fibrotic walls, lined by epithelia with the features of airway epithelia [7].

These alterations lead to a reduced lung compliance as a result of both the modification of extracellular matrix composition and the alteration of the pulmonary surfactant production [7,8]. The reduction in lung compliance is an early event in IPF and may be correlated with the degree of fibrosis, measured with vital capacity (VC) and total lung capacity (TLC), and is typically associated with a restrictive functional pattern characterized by a reduction in static (TLC) and operating (VC) lung volumes [9]. Moreover, the reduced compliance is associated with clinical features of IPF such as dyspnea, mainly sustained by the increased breathing work [7].

Another cornerstone in IPF pathophysiology is the alteration of pulmonary gas exchange: in IPF, pulmonary arteries develop intimal fibrosis, medial hypertrophy, and thickening, and veins and venules are reduced in caliber due to intimal fibrosis. Moreover, up to 65% of patients present pulmonary venule occlusion and alveolar capillary multiplication and muscularization [7]. These pathological alterations, together with interstitial thickening, cause an impairment in gas diffusion capacity and chronic arterial hypoxemia. The reduced diffusion capacity is measured with the diffusion capacity of the lung for carbon monoxide (DLCO) and it is reduced in nearly all IPF patients at the first evaluation, independently from static or operating lung volumes [10].

### 1.1. Role of Immune System

As demonstrated in animal models, inflammation precedes the development of a pro-fibrotic response, and the inflammatory alveolitis reduces the subsequent fibrotic response [11]. Several inflammatory cells have been demonstrated to be implicated in the early pulmonary fibrosis in animal models [12].

Macrophages are mainly involved in the maintenance of homeostasis and resistance to invasion by pathogens [13]. They undergo M1 (classical) or M2 (alternative) activation depending on which stimuli they are exposed to: classical M1 activation is stimulated by toll-like receptor ligands and INF-y, whereas alternative M2 activation is stimulated by interleukin-4 (IL-4) and IL-13 [14].

The M1 phenotype is characterized by the expression of proinflammatory cytokines, the production of reactive nitrogen and oxygen intermediates, the promotion of a T-helper 1 (Th1) response, and microbiological and tumoricidal activity; on the other hand, M2 macrophages are involved in parasite containment and the promotion of tissue remodeling, tumor progression, and immunoregulation. M2 is characterized by phagocytic activity and the expression of scavenging molecules, mannose, and galactose receptors [14]. Macrophage polarization is a dynamic process that can be reproduced in vitro [15,16] and it has been demonstrated that several diseases reflect changes in macrophage activation, with classically activated M1 cells implicated in starting and continuing inflammation, while M2 macrophages are associated with the resolution or smouldering of chronic inflammation [17].

In IPF, M1 macrophages contribute to the host defense by generating reactive nitric oxide and releasing proinflammatory cytokines and chemokines that have a strong antimicrobial and anti-tumoral activity [18]. According to the current pathogenetic model of IPF, lung fibrosis is the final pathological outcome of aberrant healing responses to persistent lung injury. Pulmonary cellular damage induced by several factors (environmental particulates, infections, mechanical damage) results in the disruption of lung parenchymal architecture. At the early inflammatory stages, acute lung injury promotes an M1 phenotype under the control of interferon regulatory factor 5 (IRF-5), with the expression of high levels of inducible nitric oxide synthase (iNOS) and proinflammatory cytokines; the persistent and sustained inflammatory responses act as a trigger to initiate the fibrotic response in the lung [18].

M2 polarization can be induced by several mediators such as IL-4, IL-13, Transforming Growth Factor β (TGF-β), and/or IL-10, all of which are implicated in the wound-healing fibrosis cascade [19]. During the development and progression of IPF, the predominant infiltration of M2 macrophages in fibrotic areas acts as a key regulator of fibrogenesis; in particular, M2 macrophages produce profibrotic mediators such as TGF-β and Platelet-Derived Growth Factor (PDGF) by which they induce continuous fibroblast activation and promote myofibroblast proliferation. Moreover, IL-10 generates a Th2 microenvironment that involves fibrocyte recruitment and M2 macrophage activation, leading to excessive extracellular matrix deposition [18].

The role of lymphocytes in IPF is still poorly understood and controversial; indeed, lymphocyte-modulating therapies are ineffective against IPF and lymphocytes are not required for the development of induced fibrosis in mice models [20]. Nevertheless, lymphocytes may contribute to fibrosis through poorly defined mechanisms. T lymphocytes have been identified in BAL fluid and lung parenchyma of IPF patients [21], but the majority of studies have been conducted on peripheral blood samples. Furthermore, contradictory results have been published on some subsets of lymphocytes, i.e., regulatory T cells have been found either increased or reduced in patients with IPF [2]. Generally, it was demonstrated that changes in the proportion or activation of some T and B cell subsets, as well as the presence of specific autoantibodies, would negatively influence the progression of the disease, accelerating the lung function decline [2].

### 1.2. Viruses

During recent years, many viruses have been claimed as putative actors in the development of IPF. Cytomegalovirus was demonstrated to accelerate existing fibrosis in bleomycin-model mice by enhancing TGF-β1 activation and by increasing vimentin levels [22]. Epstein–Barr virus (EBV) was proposed to be associated with IPF development: Stewart et al. demonstrated that EBV DNA was present in more than 40% of IPF patients’ lung biopsies [23] and, subsequently, Manika et al. found significantly higher EBV DNA copies in the BAL fluid of IPF patients than in subjects with other interstitial lung diseases [24]. Although the mechanism is still not completely understood, EBV pro-fibrotic damage is carried out by acting through the upregulation of TGF-β1 expression and inducing mesenchymal characteristics in epithelial cells [25]. However, other studies did not confirm these results, raising many doubts on the real magnitude of the role of EBV and other herpes viruses on IPF development [26]. Finally, no studies have been conducted on the role of other viruses in IPF patients and no studies have reported possible effects of other coronaviruses than SARS-CoV-2 on the development of lung fibrosis.

### 1.3. Acute Exacerbations

Some IPF patients have a progressive and slow course of the disease while others have a rapid progression; in some cases, an acute decline in lung function, called “acute exacerbation” (AE) of IPF (AE-IPF), can be observed. Acute exacerbation is defined as an acute, clinically significant respiratory deterioration characterized by the evidence of new widespread alveolar abnormalities [27]. The diagnosis requires a known IPF and an acute clinical worsening, with a chest CT scan revealing bilateral ground glass abnormalities and/or consolidations superimposed on an UIP pneumonia [27].

Although the pathophysiology of AE-IPF is still not well understood, AE has some clinical aspects in common with acute respiratory distress syndrome (ARDS): increased oxygen needs, bilateral ground glass opacities and/or consolidations at chest CT, and diffuse alveolar damage on lung biopsies [27,28]. The pathophysiology of AE-IPF involves chronic factors, such as epithelial cell dysfunction, the accumulation and activation of fibroblasts, and acute factors i.e., acute lung injury. The combination of these two types of injuries unleashes the pro-fibrotic cascade that leads to lung injury without hyaline membrane formation and extracellular matrix deposition [27,29].

Many causes have been attributed to the development of an AE-IPF including infections, gastric aspiration, surgery, advanced lung disease, and coronaropathies [27]. Among these, viral infections in AE-IPF have been detected with pan-viral arrays and polymerase chain reaction (PCR) panels [30,31,32,33,34]. In particular, Wotton et al. detected common respiratory viruses on BAL fluid in 9% of samples obtained from AE-IPF: two cases of rhinovirus, one human coronavirus-OC43, and one parainfluenza virus-1; when the authors performed sensitive genome-specific PCR for Herpes Simplex Virus (HSV), EBV, and torque teno virus (TTV), they identified 15 additional positives (12 TTV, 2 EBV, 1 HSV) [32]. Moreover, in autoptic studies including patients who died from AE-IPF, authors identified occult respiratory infection [28,35].

Even though many efforts have been made to better understand the pathogenesis and to develop new antifibrotic agents able to reduce the incidence of AE events [36,37], AE-IPF prognosis is still poor and precedes nearly 40% of IPF deaths, with a median survival following an AE of three to four months [27].

## 2. COVID-19 and Fibrotic Damage

### 2.1. Lung Fibrosis

Since the first coronavirus disease 2019 (COVID-19) wave, many clinicians speculated about the possibility of the development of a post-COVID19 pulmonary fibrosis. Even if post-viral fibrosis is a rare condition, due to the massive scale of the COVID-19 pandemic, the magnitude of the problem could have become non-negligible.

When we looked for other past coronavirus infections, some studies evaluated the pulmonary function of patients who survived severe acute respiratory syndrome (SARS), and the long-term evaluation revealed a reduced lung function in 28% to 53% of patients, who reported symptoms like those of patients with pulmonary fibrotic diseases [38].

Pulmonary fibrosis is a complex process, involving many pathways. Alveolar epithelial cells (AECs) are involved in the regulation of the inflammatory response upon environmental acute injury, and AEC type I (AEC1) and AEC type II (AEC2) have different roles and functions. AEC1 represents the squamous layer surrounding the alveolar airspace, being involved in gas exchanges [38]; AEC2 produces surfactants and it is the predominant epithelial progenitor cell that can differentiate into AEC1 cells. The reduction in AEC amount or the loss of their function can lead to an improper repair of the lung parenchyma and to lung fibrosis [38,39].

A variety of intrinsic and extrinsic causes can activate the pro-fibrotic cascade, and COVID-19 can be one of these factors. After the clearance of the insult by the immune system, AEC2 proliferates to repair the damage; if the insult causes a dysfunction of AEC2, these cells recruit myofibroblasts that lead to the activation of collagen synthesis, apoptosis of AEC2, and, finally, the onset of pulmonary fibrosis. The longer the insult acts on lung tissue, the lower is the probability of a complete resolution of the damage [38].

The effectors of the upregulation of the inflammation process are cytokines and the major stimulator is TGF-β, mainly secreted by alveolar macrophages, bronchial epithelial cells, and hyperplastic AEC2 in response to alveolar damage and inflammation [40]. TGF-β exerts its pro-fibrotic activity, stimulating several pathways, but the most important are the modulation of extracellular matrix deposition by the activation of myofibroblasts, as well as the activation of P120-catenin to increase fibroblast foci [38].

Myofibroblasts derive from fibroblasts and are recruited to facilitate the repair process; they migrate and secrete extracellular matrix, stabilizing the repair process and circumscribing the inflammatory course. The imbalance of extracellular matrix deposition generates the fibrosis [38].

Finally, reactive oxygen species (ROS), when produced in excess from the innate inflammatory system, induce epithelial apoptosis and the secretion of profibrotic cytokines, and stimulate the differentiation of fibroblasts in myofibroblasts (Figure 1) [41].

### 2.2. COVID-19-Mediated Lung Fibrosis

SARS-CoV-2 is an enveloped, single-stranded, positive-sense RNA virus. SARS-CoV-2 is the causative agent of COVID-19, and the primary mode of inter-human transmission has been identified to be mainly via respiratory droplets [42,43,44]. During these three years of the pandemic, several treatments have been proposed, demonstrated to be effective in improving patients’ outcomes and then authorized for treatment [45].

The main targets of SARS-CoV-2 infection are: AEC2 cells, alveolar macrophages, basal epithelial cells of the upper airways, and enterocytes [38]. As specified above, AEC2 cells and alveolar macrophages are two of the main actors in the development of lung fibrosis. Moreover, COVID-19 patients presented high levels of: (1) INF-γ, known for increasing the Th1 response and a profibrotic inflammatory profile [38,46]; (2) TGF-β, which enhances the recruitment of fibroblasts with the subsequent differentiation in myofibroblasts [38]; (3) IL-17, which stimulates neutrophil degranulation and oxidative stress damage, both involved in the development of fibrosis [47] (Figure 2).

But what is the real magnitude of the post-COVID-19 pulmonary fibrosis (PCPF)? A recent meta-analysis by Amin et al. tried to address this question. The authors included 13 studies with more than two thousand patients. The prevalence of PCPF was 44.9% and the radiological assessment was variable among studies and ranged from the hospital discharge day to 7 months after. The prevalence was higher among males (53.8% vs. 46.2%) and among patients with a history of chronic obstructive pulmonary disease (COPD, 2.88 times higher). Compared to non-fibrotic patients, those with PCPF had more common respiratory symptoms, such as cough (47.4%) and chest pain (27.6%) during the COVID-19 acute phase, as well a more persistent dyspnea, cough, chest pain, and myalgia during the follow-up [48]. The main criticism is the heterogeneity of studies included in the analysis: there is no widely accepted definition for PCPF, with respect to neither radiological nor functional parameters; moreover, there is no general agreement about the standardized follow-up period that can define the stabilization of pulmonary CT alterations. The most common CT alterations are represented by parenchymal bands, interlobular septal thickening, and reticulations [49]. We know neither the clinical significance of these alterations nor their development during the subsequent follow-up. We know that most CT alterations present during the acute phase of the disease decrease during the follow-up without a significant correlation between CT severity lung scores and spirometric results obtained up to 18 months after discharge [50].

## 3. COVID-19 and IPF: Similarities

After all these considerations, there are several links between the PCLF and IPF. First, all the mediators that are involved in lung fibrosis after SARS-CoV-2 infection are also relevant in the development of fibrosis in IPF patients: this parallelism, even if not yet supported by animal models, or in vitro or in vivo studies, strengthens their role in PCLF development [38]. Another factor that connects COVID-19 and IPF is a genetic overlap: recently, Allen et al. demonstrated a positive genome-wide genetic correlation between IPF and severe COVID-19 risk using large genome-wide association studies (GWASs); they showed for the first time the same underlying causal variant to the genetic association signals of *MUC5B*, *DPP9*, and *ATP11A* [51].

Sinha et al. analyzed more than a thousand human lung transcriptomic datasets, focusing on IPF-specific signatures, AEC2 cytopathies, prognostic monocyte-driven processes that are known to be correlated to IPF development, and one COVID-derived signature. The authors found that COVID-19 and IPF show similar gene expression patterns, cytokine storms, and AEC2 cytopathic changes. Moreover, Sinha et al. demonstrated that these immunocytopathic features were reversible with anti-SARS-CoV-2 therapies in a hamster model [52].

Finally, several clinical experiences in the treatment of PCLF with nintedanib are rising. A possible role of antifibrotic therapy in PCLF was initially supposed by George et al.: they speculated about a role of antifibrotics in the prevention of lung injury because these treatments have rapid effects and benefits in other forms of lung fibrosis triggered by viral infection. Secondly, these treatments have the TGF-β pathway as the main target, which is one of the COVID-19-mediated pro-fibrotic stimuli. Other targets could be the αvβ6 integrin, galectin, and PLN-74809, all involved in the SARS-CoV-2 proinflammatory pathway. Ultimately, rapamycin and PI3K inhibitors have shown promising results in IPF [53].

Nintedanib is a tyrosine kinase inhibitor that was approved as an antifibrotic drug for the treatment of IPF [54] and was demonstrated to be effective in reducing the progression of IPF measured by the decline in FVC [55]. After the first case report [56], several articles reported the use of nintedanib in the treatment of pulmonary fibrosis in severe pneumonia induced by COVID-19. Umemura et al. compared the treatment of 30 patients to an historical cohort of patients: they did not report significant differences in the 28-day mortality (23% vs. 20%), but the length of mechanical ventilation was shorter and the chest CT alteration was lower in the nintedanib-treated group [57]. In addition, Saiphoklang et al. demonstrated that Nintedanib, added to conventional anti-SARS-CoV-2 therapies (antivirals, steroids, antibiotics, anticoagulants, and immunomodulatory agents), in the acute COVID-19 phase, did not improve oxygenation, chest X-ray findings, or the 60-day mortality, but improved the SpO_2_/FiO_2_ ratio in treated patients [58].

## 4. IPF and COVID-19 in Clinical Practice

During the COVID-19 outbreak, patients affected by ILD have been considered to be at high risk to evolve into severe COVID-19. This condition was sustained by the propensity to develop the AE of lung fibrosis, altered lung function, and ongoing treatment with immunomodulatory effects that could reduce the viral clearance [53,59].

COVID-19 was demonstrated to be a trigger for AE IPF: SARS-CoV-2 induces a cytokine storm in severe cases, with an acute escalation of inflammation in lung parenchyma. However, it is difficult to distinguish between severe COVID-19 from AE IPF triggered by COVID-19 because clinical and radiological manifestations are similar [60]. Moreover, due to the widespread use of the CT scan during the COVID-19 pandemic, we diagnosed many interstitial lung diseases during the acute phase of COVID-19 disease, making it difficult to distinguish an AE of the chronic ILD triggered by SARS-CoV-2 infection, from SARS-CoV-2 pneumonia superimposed on a pre-existing ILD.

As supposed, it has been observed that IPF patients represented a high-risk cohort for poor outcomes when infected by SARS-CoV-2. Indeed, this population was more susceptible to SARS-CoV-2 infection: a national Korean cohort of COVID-19 patients demonstrated that the proportion of patients with an underlying ILD was significantly higher in the COVID-19 cohort than in the non-COVID-19 control cohort (0.8% vs. 0.4%): this indicates that patients affected by ILDs were more susceptible to SARS-CoV-2 infection with an odds ratio of 2.02 [61].

IPF patients had a more severe disease as suggested by Esposito et al. who first reported an increased mortality rate among patients with ILD (33% with an odds ratio of 3.2) [62]. Later, Naqvi et al. confirmed that IPF patients had a higher risk of mortality, critical care admission, and need for mechanical ventilation 30 days and 60 days from COVID-19 diagnosis. In particular, 30-day and 60-day mortalities were, respectively, 15.94% and 18.33%. IPF patients were seven timed more likely to die or be on mechanical ventilation at 30 days [63].

A recent meta-analysis included 15 studies with 135,263 COVID-19 patients: the authors observed a prevalence of comorbid ILD amounting to 1.4%; the prevalence of ILD in non-survival patients with SARS-CoV-2 infection was 2.7-fold higher. The mortality and intensive care unit admission among patients with ILD were, respectively, 2.4 and 3.0 times higher than those of patients without ILDs [64].

Finally, it could be interesting to consider some non-clinical problems that were raised during the pandemic: one above all, the accessibility to medical structures for chronic patients was limited. The majority of hospitals significantly reduced the number of outpatient accesses, reducing the number of visits. This certainly negatively influenced the outcomes of chronic patients: this could be derived by Faverio et al.’s study where they reported a significant increase in mortality in IPF patients during the post-lockdown period, not apparently correlated to COVID-19 [65].

## 5. Post-COVID-19 Respiratory Functional Status Evaluation

Spirometry, diffusion capacity, and lung volume evaluations are the most standardized and reproducible measurements of lung function. International guidelines suggest that patients who underwent severe complications of COVID-19, mainly severe pneumonia or acute respiratory distress syndrome (ARDS), should undergo complete pulmonary functional tests (PFTs) within 12 weeks after hospital discharge [66]. In the case of mild to moderate pneumonia, guidelines suggest that PFTs should be performed in the case of persistent abnormalities at radiological evaluation. Finally, the patients should be referred to a pneumological evaluation in the case of altered PFTs with abnormalities at the chest CT [66].

The most common alteration of PFTs during the post-COVID-19 follow-up was related to DLCO. In particular, a recent meta-analysis showed that 39% of patients had a reduced DLCO [67]. The majority of studies reported a mild reduction in DLCO at first post-acute evaluation (from two weeks up to 3 months). Studies that included patients evaluated more than 2 months after the hospital discharge evidenced that those who required ventilatory support (non-invasive or invasive mechanical ventilation, high-flow oxygen therapy) had a risk 4.6 times higher of having an impaired DLCO with respect to those who did not require oxygen implementation during hospitalization [68]. Other factors that have been associated with a reduction in DLCO are female gender, the presence of chronic kidney failure, the modality of oxygen delivery (non-invasive ventilation), and intensive care unit admission [69]. Moreover, DLCO correlated with the clinical, radiological, and humoral impairment of COVID-19 patients [68]; Quin et al. demonstrated that chest CT total severity score was significantly associated with a reduced DLCO [70]. When DLCO was longitudinally evaluated, it did not significantly change over time, with a persistent impairment during the one-year follow-up [71]. As specified above, DLCO measures the gas-exchange capacity of the alveolar-capillary barrier and it is the result of the carbon monoxide transfer coefficient (KCO) and alveolar volume (AV) multiplication. KCO depends on the thickness and area of the alveolar capillary membrane, the blood volume in capillaries of ventilated alveoli, and the hemoglobin concentration in alveolar capillary blood [72]. COVID-19-mediated lungs damage the interests of both alveolar epithelial and endothelial cells and both can contribute to DLCO impairment. This results in a difficult interpretation of which is the predominant factor [72].

Concerning the spirometry and lung volume measurement, the majority of studies reported a normal FVC, forced expiratory volume in the first second (FEV1), and FEV1/FVC ratio [67]; the short-term post-COVID-19 prevalence of the restrictive pattern was 15% [67]. When patients were prospectively evaluated, a restrictive abnormality (defined as FVC < 80% of the predicted value) was present in 14%, 9%, and 6% of patients at 2, 6, and 12 months, respectively, but the mean FVC (% of the predicted value) was nearly 100% in all longitudinal measurements [73].

## 6. Post-COVID-19 Follow-Up

As specified above, there is no standardized follow-up scheme in post-COVID-19 patients. The optimal timing for follow-up clinical and functional evaluation is unknown and it depends on many factors, mainly related to the persistence of symptoms, the severity of COVID-19, and resource availability.

The European Respiratory Society (ERS) published a statement on Long COVID-19 follow-up: long COVID-19 is a condition that describes signs and symptoms that develop after an infection consistent with COVID-19, which continues for more than 12 weeks and cannot be explained by an alternative diagnosis [74]. The patients’ referral for a post-COVID-19 evaluation is recommended when symptoms persist for 6–12 weeks [74]. European respiratory societies proposed different algorithms for evaluating COVID-19 survivors: the British Thoracic Society (BTS) suggested an evaluation in the first three months after hospital discharge depending on the severity of COVID-19 infection and ICU admission; in patients with severe or mild-to-moderate COVID-19, a clinical and cardiopulmonary evaluation was recommended at 12 weeks [75].

PFTs with the measurement of TLC and DLCO and a six-minute walking test (6MWT), together with cardiopulmonary exercise testing, have been the most diffused post-COVID-19 functional evaluation [74]. The high heterogeneity among studies makes it difficult to draw definitive conclusions: the majority of studies involved hospitalized patients, but also patients with pre-existing chronic lung diseases were included in the analysis, leading to functional parameter reports that were not standardized [74].

Most investigators concluded that PFTs including static lung volume measurements (at least TLC), expiratory flows, and DLCO evaluations were a useful tool to assess long-term function sequelae in COVID-19 patients [74].

Follow-up imaging of post-COVID-19 patients is one of the most debated topics in the ERS statement. Many studies reported pulmonary abnormalities at the three- and six-month post-infection CT scan: these radiological abnormalities at three to six months may overestimate the real frequency of persistent alterations, which are more frequently observed in patients who had a more severe acute phase with extensive imaging abnormalities [74]. CT alterations reduced their extension over time [76], but fibrotic-like changes (i.e., parenchymal bands, traction bronchiectasis) may be found in a consistent percentage of patients at the three- to six-month follow-up even if it is still unclear whether these alterations represent irreversible disease, a slowly progressive fibrotic disease, a minimal distortion of lung architecture stable during subsequent years, or the future disappearance [74]. For these reasons, ERS task force members are cautious when calling out fibrosis, and some of them recommend repeating a CT at 12 weeks post-discharge in patients with persistent symptoms [74,75].

Based on these recommendations and on our most recent published studies [69,71], we suggest a follow-up visit at 4 months from acute SARS-CoV-2 infection with a complete clinical and respiratory functional evaluation (measurement of TLC and DLCO). In the case of restrictive patterns (TLC < 80%), an impairment of DLCO, and persistence of clinical symptoms, we suggest the repetition of an HRCT scan: in the case of fibrotic features (reticulations, traction bronchiectasis, persistent parenchymal consolidations) involving more than 10% of the lung parenchyma, we suggest a specialistic interstitial lung disease pulmonology evaluation to continue the clinical and functional follow-up; in the case of an absence of fibrotic features at CT, a clinical and functional evaluation (in addition to other investigations based on clinical and radiological results) after 6 months is suggested (Figure 3).

## 7. Antifibrotic Agents Used in IPF for the Treatment of Post-COVID-19 Pulmonary Fibrosis: Nintedanib and Pirfenidone

Since the hypothesis that COVID-19 could lead to the development of a PCPF, several drugs have been purposed for the treatment [53]. Up to now, there are many ongoing clinical trials that are evaluating different molecules on PCLF.

As specified above, nintedanib is an antifibrotic drug that inhibits pro-fibrotic mediators, platelet-derived growth factor (PDGF), and fibroblast growth factor (FGF), binding the intracellular ATP pockets of the corresponding receptors and therefore inhibiting the pro-fibrotic signaling. This activity reduces the proliferation, migration, and differentiation of myofibroblasts and then the deposition of extracellular matrix [77]. Pirfenidone is an antifibrotic agent that acts by regulating pro-fibrotic and pro-inflammatory cytokine cascades and inhibiting the proliferation of fibroblasts and collagen synthesis [78].

In August 2020, the FIBRO-COVID trial started to recruit patients with fibrotic changes after COVID-19: inclusion criteria were: age older than 18 years, previous SARS-CoV-2 infection complicated with severe pneumonia and ARDS, and clinical and radiological signs of pulmonary fibrosis at chest HRCT (fibrotic changes no less than 5% after recovery). The aim of the study was to evaluate the effects of pirfenidone on FVC and HRCT changes after 24 weeks of treatment in a placebo-controlled trial. The trial is still recruiting patients and no results have been published yet [79].

The first trial assessing nintedanib use in PCLF started in April 2022: the study, a randomized placebo-controlled trial, aims to evaluate the efficacy and safety of nintedanib in the treatment of PCLF in patients with moderate to severe COVID symptoms. In particular, the primary and secondary outcomes of the study are the changes in FVC, DLCO, and 6MWT distance and fibrotic alterations at HRCT after 8 weeks of treatments. The recruitment status of this trial is unknown, as well as preliminary results [80].

The NINTECOR trial is the second study that evaluates nintedanib compared to placebo; inclusion criteria were: 18-89-years-old patients with a history of COVID-19, radiological CT signs of fibrosis including more than 10% of the lung and DLCO below 70% of the predicted value. The authors aim to evaluate respiratory functional status (FVC, DLCO) changes, exercise tolerance, pulmonary fibrotic modifications at HRCT, quality of life, dyspnea, anxiety and depression levels, and other bio-humoral evaluations 12 months after the introduction of nintedanib or placebo. This study is still recruiting patients and no preliminary results have been published up to now [81].

Finally, the phase IV ENDCOV-I clinical trial, active since November 2020, aims to evaluate the influence of nintedanib on slowing the fibrotic damage in patients with lung fibrotic alterations on a chest X-ray or chest HRCT, FVC < 80%, and DLCO < 50% of predicted values, not less than 4 weeks after the emergence of the first symptoms. Patients will receive nintedanib for 6 months and the functional and clinical status as well as radiological alterations will be evaluated prospectively. The study is still ongoing, and patients’ recruitment is open [82].

The main criticism of these studies is that there is no standardized definition of PCLF. Due to the limited observations present in the literature, there is no uniform temporal, functional, or radiological, or the association of all these features, definition for fibrotic damage. The fibrotic definition of the pirfenidone trial was a HRCT with fibrotic radiological changes of at least 5% after recovery from the acute process. In nintedanib trials, NINTECOR defined PCLF as those cases with a radiological extension >10% (with fibrotic features) and DLCO ≤ 70% of the predicted value, whereas the ENDCOV-I trial did not define the percentage of fibrotic CT extension but included FVC in the diagnosis (FVC < 90% and DLCO < 70%). None of the nintedanib trials included in the clinical definition of PCLF showed the presence of symptoms consistent with a pulmonary fibrotic disease (i.e., exertion dyspnea, dry cough), but both NINTECOR and ENDCOV-I included clinical outcomes as secondary outcomes.

## 8. Conclusions

In conclusion, there are several links between the paradigmatic example of lung fibrosis, IPF, and the rising evidence of a post-COVID-19 fibrotic disease. Genetic, pathological, and clinical similarities have been demonstrated as well as positive results in the treatment of COVID-19 with antifibrotic drugs. It is still unclear which is the real magnitude of the PCPF, and an international-wide definition is needed to standardize diagnostic criteria.

## Figures and Tables

**Figure 1 microorganisms-11-00895-f001:**
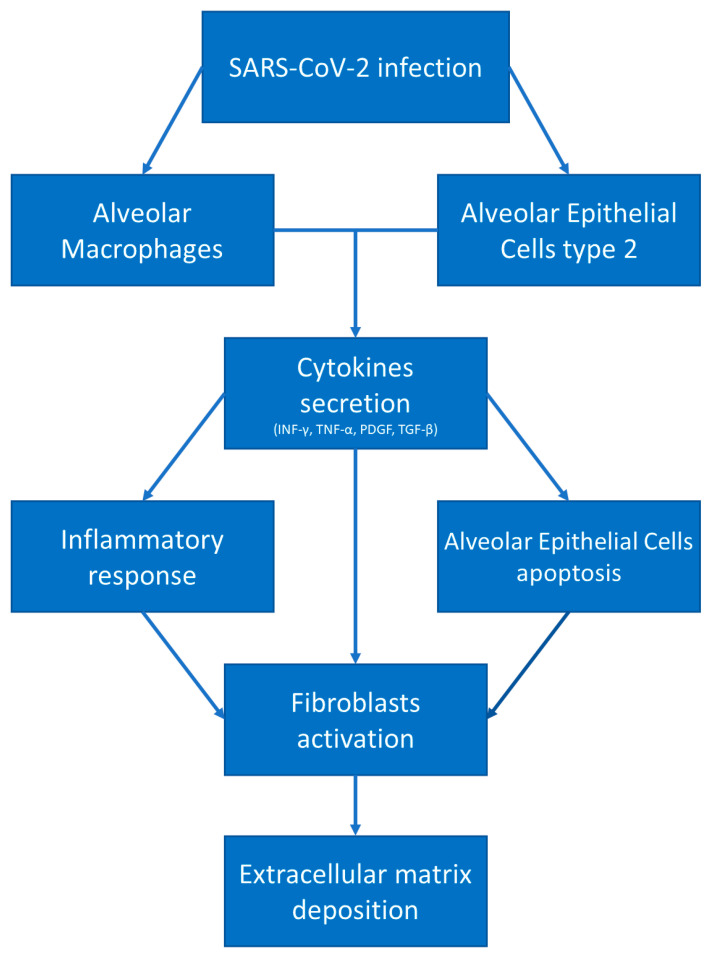
COVID-19 and fibrotic damage. After SARS-CoV-2 infection, alveolar macrophages and alveolar epithelial cells type 2 secrete pro-inflammatory and pro-fibrotic cytokines, leading to fibroblast activation and differentiation in myofibroblasts, resulting in extracellular matrix deposition.

**Figure 2 microorganisms-11-00895-f002:**
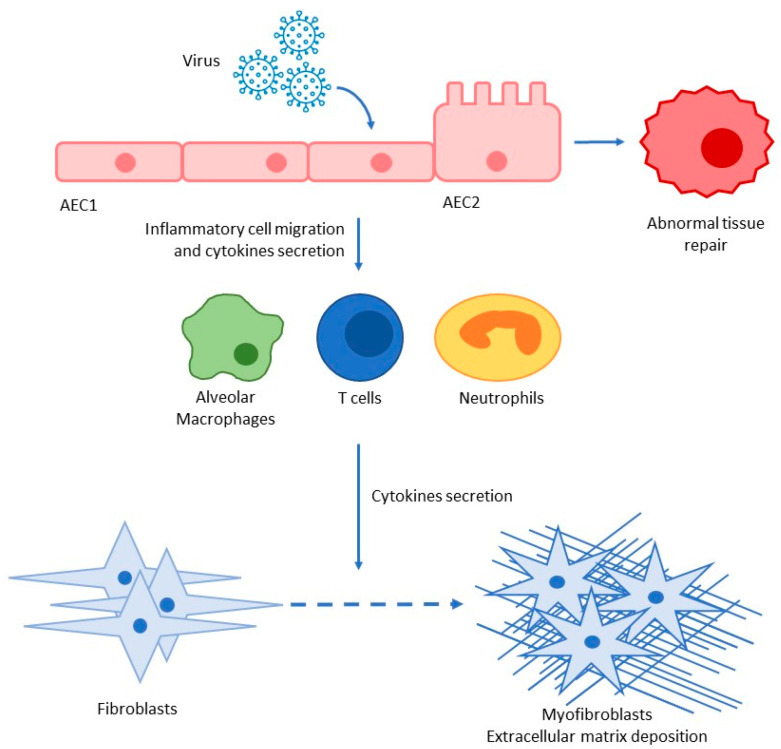
Pro-inflammatory and pro-fibrotic cascade after SARS-CoV-2 infection; AEC alveolar epithelial cells type I and II.

**Figure 3 microorganisms-11-00895-f003:**
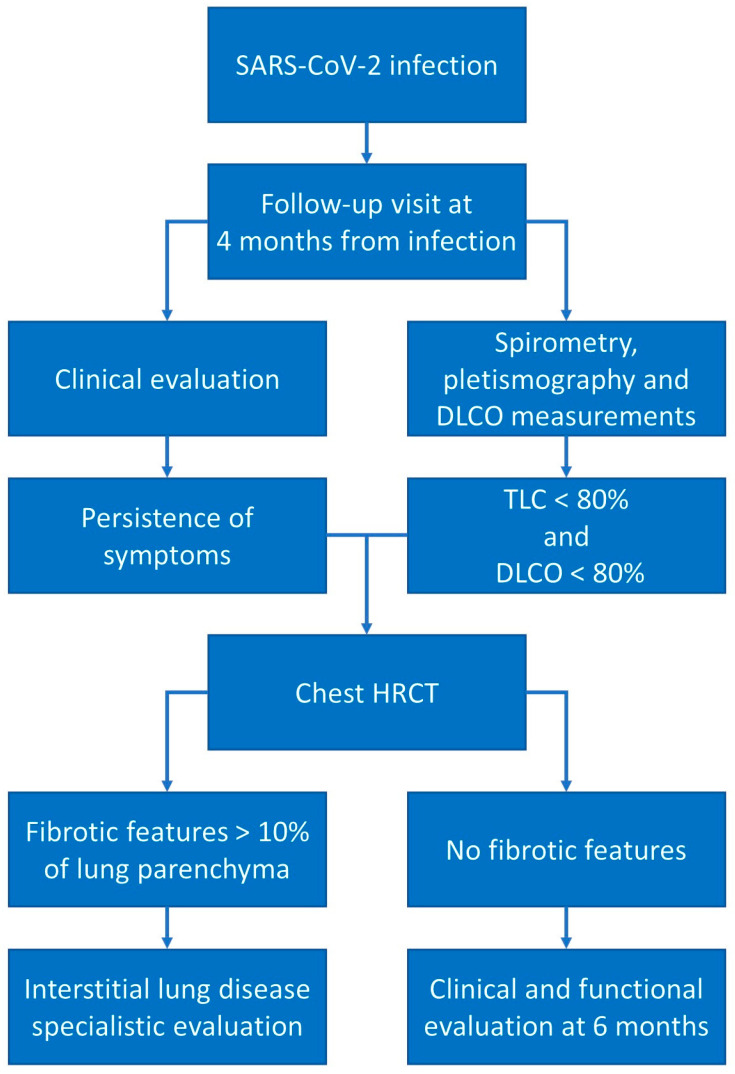
Post-COVID-19 follow-up schemes. In the presence of clinical and respiratory functional alterations, we suggest a follow-up high-resolution CT scan to investigate the presence of fibrotic alterations.

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
