# Peer review of "Idiopathic Pulmonary Fibrosis and Post-COVID-19 Lung Fibrosis: Links and Risks"

_microorganisms, 2023, doi:10.3390/microorganisms11040895_

Round 1

Reviewer 1 Report

The review  ”Idiopathic Pulmonary Fibrosis and post-COVID-19 lung fibro-2 sis: links and risks” by Filippo Patrucco and his colleagues is an interesting topic and covers many aspects of the subject. However, I have two concerns regarding this manuscript.

1) Please carefully check the language of the whole manuscript. There are many mistakes. For example, in the Abstract part,

1.      line 18: change  “mechanisms” to “mechanism”

2.      line 19: change “orchestrate” to “orchestrating”

3.      line 22: change “acute phase of SARS-CoV-2 infection” to “acute phase caused by SARS-CoV-2 infection”

4.      line 24: “physio pathological intra ed extracellular”?

5.      line 25:  change “antifibrotic treatments used in fibrotic progressive diseases” to “antifibrotic treatments for fibrotic diseases”

6.      line 26 change “the cause of acute exacerbation” to “a cause of acute exacerbation”;  change “ be responsible of” to “be responsible for”

7.      line 29 change “among” to “between”

2) Please summarize the findings of literatures comprehensively and precisely, and also cite literature properly. For example:

line 132: “Epstein-Barr virus (EBV) was found significantly higher in BAL fluid and lung tissue of…” Does that mean the amount of EBV?  From the literatures, it seems that the authors would like to say “A higher occurrence of Epstein-Barr virus (EBV) in BAL fluid and the lung tissue of patients with IPF.

In addition, the other original paper (JAMES P. STEWART, 1998, PubMed: 10194186 ) should be cited.

Also, other associated papers should be considered. For example, Amir Hossein Jafarian and coworkers (PMID: 32095147) reported there was no statistically significant difference in the prevalence of EBV and HHV-8 DNA in the IPF specimens and controls in their study.  

Author Response

Thank you for the suggestions.

We corrected the errors that you evidenced, moreover we made an extensive revision of the English language. We added to the references list the paper that have been suggested. 

Reviewer 2 Report

The manuscript brings an interesting and comprehesive review about lung fibrosis and trace parallels between the fibrosis observed in IPF and  those triggered by severe COVID-19 forms.

While well-writen, I believe the authors should develop further the illustrations in the manuscript. Those are of major relevance to aid the readership to go through the dense content of the text. illustrations regarding cellular mechanisms involved in the fibrosis, as well as the targets of the antifrotic agents discussed in paper should be provided. Tissue micrography and other tools could also help.

Author Response

Thank you for the appreciations. We provided one more image that could better explain cellular mechanisms involved in lung fibrosis.

Reviewer 3 Report

This narrative review on the links between idiopathic and post-COVID fibrosis is interesting although the absence of a clear definition of post-COVID fibrosis is an important limitation.

The review is well written although there are a few typo or syntax errors. However, the paragraph 1.1 on the immune system and fibrosis is difficult to read. A figure with the pathways involved in idiopathic and COVID fibrosis would be more informative. The text could be reduced to explain common pathways. 

There is no definition of COVID-19 mediated fibrosis. However, a table summarizing the diagnostic variables in favor of fibrosis post COVID would be helpful. Clearly, respiratory function variables are not helpful in patients with COVID to distinguish between alveolar and intersticial damage. What is the value of BAL for the diagnosis of post COVID fibrosis ?

Author Response

Thank you for the comments. We provided an extensive English review made by a mother language. Moreover, as you requested, we added an image explaining fibrotic mechanisms. The value of BAL was not discussed because it could be misleading. We would not send the message that BAL is influent in the diagnosis of PCLF; we prefered to add the image that explain the mechanisms involved in fibrosis. We recently published an article about the results of BAL during COVID-19 infection but up to now no studies explored this evaluation in PCLF. 

Round 2

Reviewer 1 Report

Please address the concerns point by point.

I do not think that you addressed the previous concern.

2) Please summarize the findings of the literature comprehensively and precisely, and also cite the literature properly. For example: (this is only one example)

line 132: “Epstein-Barr virus (EBV) was found significantly higher in BAL fluid and lung tissue of…” Does that mean the amount of EBV?  From the literature, it seems that authors would like to say “A higher occurrence of Epstein-Barr virus (EBV) in BAL fluid and the lung tissue of patients with IPF.

In addition, the other original paper (JAMES P. STEWART, 1998, PubMed: 10194186 ) should be cited.

Also, other associated papers should be considered. For example, Amir Hossein Jafarian and coworkers (PMID: 32095147) reported there was no statistically significant difference in the prevalence of EBV and HHV-8 DNA in the IPF specimens and controls in their study. 

Author Response

Thank you for the quick response. Your questions are significant because we did not added the changes to the text in this version that was revised by the other authors. For this reason we apologize to you. 

In this version, we specified that EBV was found on lung biopsies and BAL fluid of IPF patients, citing the two main studies published on this specific argument (Stewart and Manika); moreover, we deleted the role of IL17 (that was only marginally discussed) and we added to the text the doubts on the role of EBV and other herpes viruses in IPF development. 

These variations are visible from line 127 to 136.
